# Halotolerant Rhizobacteria for Salinity-Stress Mitigation: Diversity, Mechanisms and Molecular Approaches

Alka Sagar [1,*], Shalini Rai [2], Noshin Ilyas [3], R. Z. Sayyed [4,*], Ahmad I. Al-Turki [5], Hesham Ali El Enshasy [6,7] and Tualar Simarmata [8]

1 Department of Microbiology, Meerut Institute of Engineering and Technology [MIET], Meerut 250005, India
2 Department of Biotechnology, SHEPA, Varanasi 221011, India; shalinimicro09@gmail.com
3 Department of Botany, PMAS Arid Agriculture University, Rawalpindi 46300, Pakistan; noshinilyas@yahoo.com
4 Asian PGPR Society for Sustainable Agriculture, Auburn University, Auburn, AL 36830, USA
5 Department of Plant Production and Protection, College of Agriculture and Veterinary Medicine, Qassim University, P.O. Box 6622, Buraidah 51452, Saudi Arabia; trky@qu.edu.sa
6 Institute of Bioproduct Development (IBD), Universiti Teknologi Malaysia (UTM), Skudai 81310, Johor, Malaysia; henshasy@ibd.utm.my
7 City of Scientific Research and Technology Applications (SRTA), New Burg Al Arab, Alexandria 21934, Egypt
8 Department of Soil Sciences and Land Resources Management, Faculty of Agriculture, Universitas Padjadjaran, Jl. Raya Bandung—Sumedang km 21, Jatinangor 65363, Indonesia; tualar.simarmata@unpad.ac.id
* Correspondence: alka.sagar@miet.ac.in (A.S.); sayyedrz@gmail.com (R.Z.S.)

**Abstract:** Agriculture is the best foundation for human livelihoods, and, in this respect, crop production has been forced to adopt sustainable farming practices. However, soil salinity severely affects crop growth, the degradation of soil quality, and fertility in many countries of the world. This results in the loss of profitability, the growth of agricultural yields, and the step-by-step decline of the soil nutrient content. Thus, researchers have focused on searching for halotolerant and plant growth-promoting bacteria (PGPB) to increase soil fertility and productivity. The beneficial bacteria are frequently connected with the plant rhizosphere and can alleviate plant growth under salinity stress through direct or indirect mechanisms. In this context, PGPB have attained a unique position. The responses include an increased rate of photosynthesis, high production of antioxidants, osmolyte accumulation, decreased Na$^+$ ions, maintenance of the water balance, a high germination rate, and well-developed root and shoot elongation under salt-stress conditions. Therefore, the use of PGPB as bioformulations under salinity stress has been an emerging research avenue for the last few years, and applications of biopesticides and biofertilizers are being considered as alternative tools for sustainable agriculture, as they are ecofriendly and minimize all kinds of stresses. Halotolerant PGPB possess greater potential for use in salinity-affected soil as sustainable bioinoculants and for the bioremediation of salt-affected soil.

**Keywords:** antioxidants; bioformulation; direct or indirect mechanisms; plant growth-promoting bacteria (PGPB); salinity stress; sustainable agriculture

## 1. Introduction

The growing population and the food demand are challenging for any country's government, resulting in pressure shifts towards agricultural practices and crop productivity enhancement. Enhanced crop productivity is quite complicated in different agro-ecosystems, and is mainly affected by climatic conditions, edaphic factors, farming practices, and management techniques [1]. A series of abiotic factors comprising temperature, drought, soil pH, salinity, heavy metal, and the application of pesticides and chemical fertilizer hamper crop productivity [2]. Among all of these, salinity stress is recognized as a real threat to agricultural production [3]. In the past few years, soil salinity

and continuously changing climatic conditions have created food insecurity in several countries and have affected arid and semiarid areas of the world, at the rate of 1–2% every year [4,5]. According to some reports, more than a third of irrigated land may become barren because of the accelerated rate of salinization [6]. According to the data released by using the latest version of the FAO main report [7], soil salinity has been identified as an emerging threat to sustainable agriculture. Several recent kinds of research have estimated that around 1,128 million ha are globally aggrieved by high salinity, including 20% of the total cultivated land, and 33% of irrigated agricultural land [8]. In India, approximately 5% of the net cultivated area is salinity-affected, and that covers the Indo-Gangetic plains-lined states [9].

Zorb et al. [10] list the factors that directly influence salinity: low precipitation, higher rates of evapotranspiration, the leaching of minerals from the plant rhizosphere, anthropogenic activity, farming practices, and the formation of water-stressed conditions in plants. These factors lead to soil salinization that seriously affects the biodiversity of flora and fauna, the water quality, the water supply critical for human needs, industry, agricultural productivity, and quality, and cause a reduction in the cultivable area as well as in crops [11]. A high salt concentration contrarily strikes the physiological and biological soil processes of plants, such as the decomposition of residue, respiration, denitrification, nitrification, microbial activity, soil biodiversity, and the plant–microbe interaction [12]. The excessive application of agrochemicals and chemical fertilizers enhances the salinity input in soil and reduces crop productivity [13]. In this sequence, some agricultural practices, such as down plowing and tilling, more profoundly increase the evaporation rate of water from the soil and the deposition of salts that affect crop production. In addition, the salinity of the soil can also increase because of the salts present in the irrigation water, which reduce productivity [14]. Soil is defined as saline when its electrical conductivity (EC) exceeds 4 dSm-1 (approximately 40 mM $Na^+Cl^-$). Salinity levels are essential for reducing the yields of many crop plants [15]. Different types of salts, e.g., sodium chloride ($Na^+Cl^-$), sodium sulphate ($Na_2SO_4$), sodium nitrate ($NaNO_3$), magnesium sulphate [$MgSO_4$], magnesium chloride ($MgCl_2$), potassium sulphate ($K_2SO_4$), calcium carbonate ($CaCO_3$), etc., are present in saline soil, in which $Na^+Cl^-$ causes severe problems for higher plants [16].The mitigation of the saline effect from the soil is an exhaustive method and involves considerable time and money. However, for a long time, saline soils have been improved mainly by physical and chemical processes. Removing soluble salts from the root zone is considered in the physical process, including scraping, flushing, and leaching methods [17]. Although chemical approaches use gypsum and lime as counteracting agents, when the salt concentration is too high, these methods are not supportable and are inefficient at mitigating the salinity effects [15].

Therefore, achieving manageable crop yields is the need of the hour in saline lands, and sustainable methods should also be employed, in addition to using salinity-tolerant crops or improvement approaches, including chemical neutralizers. In the last few decades, the search for salt-tolerant plant growth-promoting bacteria [PGPB] has been accelerated in order to improve the soil fertility and the nutrient acquisition in saline soil, and to enhance the productivity of crops [18]. Their inherent reaction toward salt stress is associated with their capability to produce compatible solutes, osmoprotectants, and specific transporter channels. Salt-tolerant PGPB are now a thirst of researchers and are also being used as bioinoculants to increase crop yields, improve soil health, and protect from phytopathogens. The current review targets diversity, salt-tolerant PGPB-mediated mechanisms, and the upregulation of the regarding genes that are transferred through various methods, for example, *Agrobacterium*-mediated gene transfer having an evolutionary impact on their respective hosts (plants).Their application as bioinoculants to increase crop yields has also been targeted. In addition, approaches to salt-tolerant PGPB-mediated crop improvements, new understandings, and the productivity improvement of crops facing salinity pressure are similar discourses for future concern [19,20].

## 2. Impact of Salinity on Plant Growth and Development

Several researchers have documented the effects of soil salinity on plant enlargement and development [germination, and vegetative and reproductive enlargement], biomass production, and the grain yields of several crops [21]. High salinity has significantly reduced the leaf relative water content, the uptake of nutrients, the pigment content, the gaseous exchange attributes of the leaf, the total flavonoid, the phenolic contents, the plant growth, and the biomass production [22]. The plant organs show significant differences at the anatomical, morphological, and physiological levels, adversely affected by salinity stress. The plant usually shows dynamic enlargement, physiology, metabolic pathways, and gene expression in response to salt stress. The salinity mitigation approaches include proline accumulation within cells [23], the modulation of hormones, and the gathering of metabolic products, such as glycine betaine and polyols. These also include nitric oxide (NO), and compounds to combat the formation of reactive oxygen species (ROS). NO directly or indirectly triggers the expression of several redox-regulated genes. NO also reacts with lipid radicals, and thus inhibits lipid oxidation, exerting a protective effect by scavenging superoxide radicals and forming peroxynitrite, which other cellular processes can neutralize. NO also helps activate several antioxidant enzymes, including catalase (CAT), ascorbate or thiol-dependent peroxidase (APX), glutathione reductase [GR], and superoxide dismutase (SOD).

The plant's growth in salinity conditions is revealed at two stages: the osmotic stage (early response), and the ionic stage (late response). High salinity has harmful effects on most essential plant processes, such as disrupting photosynthesis, protein synthesis, lipid metabolism, osmotic and ionic equilibrium, and energy (Figure 1).

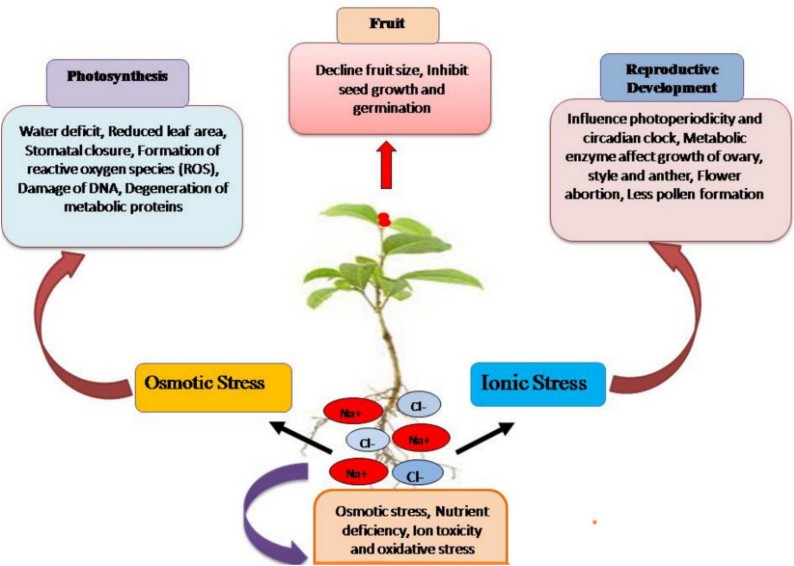

**Figure 1.** Transduction of salinity stress (osmotic and ionic stress) in plants.

The initial stage of salt stress is caused by salt present outside of the rhizospheric compartment, while the later-stage salinity stress is a consequence of the toxic effects of the salt within the plant [24]. Osmotic stress can be improved through ABA-dependent and ABA-independent pathways, whereas ionic stress is alleviated through ion homeostasis, the salt hypersensitive [SOS] pathway, and the involvement of ion transporters. Osmotic imbalance leads to water loss, reduced leaf area expansion, and stomata closure, ultimately impairing growth and the photosynthetic rate [25]. The shoot and reproductive development are affected by osmotic stress, which manifests in smaller leaves and slower emergence, and the lateral buds endure the quiescent phase and earlier flowering. The growth-modulating mechanisms use hormones and their precursors for stimulating long-distance signals from

the roots to the shoots. Phytohormone signaling triggers cell growth, the division rate, and differentiation, thus regulating the developmental morphogenesis of the plant [26].

Salinity stress in plants is a cumulative effect of osmotic and ionic stress that adversely affects plant growth and productivity. Osmotic adjustment, ion homeostasis, and detoxification are involved in the salt-stress response of plants [27]. Ionic stress develops when plants increase the $Na^+$ accumulation in their leaves above the threshold levels, leading to chlorosis and decreased photosynthesis, and other metabolic activities. Ionic stress leads to the excessive accretion of $Na^+$ in older leaves, leading to the premature aging of salt stored in grown-up leaves [28].

Various salt ions, such as bicarbonate ($HCO_3^-$), sodium ($Na^+$), potassium ($K^+$), carbonate ($CO_3^{-2}$), calcium ($Ca^{+2}$), sulfate ($SO_4^{-2}$), chloride ($Cl^-$), and magnesium ($Mg^{+2}$) cause salinization. In most salts, sodium chloride is the chief component, and its chloride ions are known to be toxic to plants and to inhibit plant growth at high levels [29]. Higher concentrations of $Na^+$ and $Cl^-$ ions in the soil affect other essential elements and reduce the plant's access needed to absorb essential nutrients and minerals [30]. Fertilizers with high-salt indexes have an osmotic effect, making it challenging to extract the water needed for plant growth. Multiple genes have been found to be associated with the salt-tolerance mechanism. A plethora of stress-related genes conferring salt-stress tolerance in plants has been reported to be involved in metabolic pathways, transcription regulation, signal transduction, and ion transporters [31].

## 3. Halotolerant/Salinity-Tolerant PGPB

Studies on halotolerant PGPB occurrence, the plant–microbe interaction, and signaling in the agricultural crop microbiome, or influence on the crop production, rank second for their copiousness, proposing in what manner the salinity is affected globally (Figure 2). The resistant breeding method is the third most relevant Halotolerant PGPB concerning the research topic, which includes developing and evaluating the mutant, screening for resistance, conventional breeding, and transgenic crops, which reduce salinity issues.

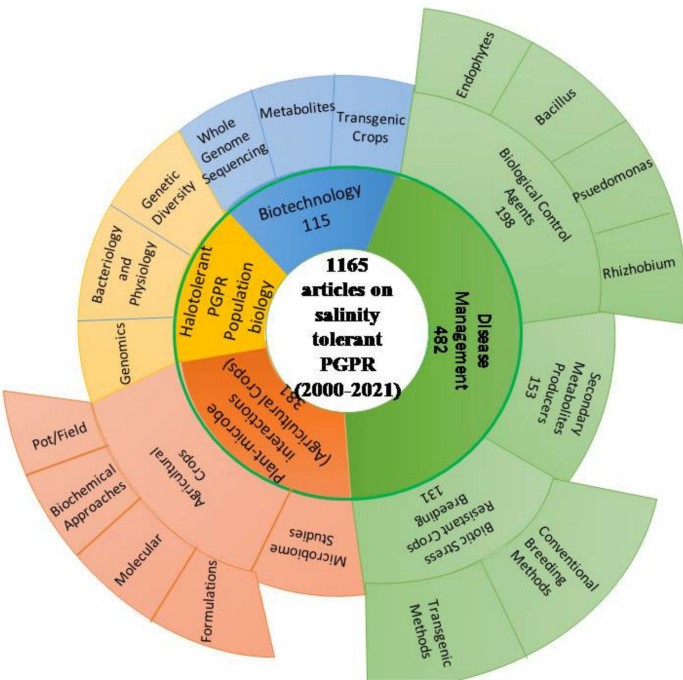

**Figure 2.** Main topics of scientific articles related to PGPB. Articles were retrieved from the PubMed database and the Google Scholar search engine (2000–2021).

Consequently, biological monitoring and the host's genetic resistance have been deliberated as the most imperative strategies for salinity-tolerant crops and disease man-

agement. In 153 papers, the naturally produced secondary metabolites of Halotolerant PGPB were studied with regard to their production, their application, and their effect on phytopathogens. The interaction of the Halotolerant PGPB with other microbes, and its population studies, have been worked on using "omics" approaches. However, most articles discuss biocontrol agents (BCAs) that comprise endophytic microbes and strains of *Bacillus* spp., *Pseudomonas* spp., and *Rhizobium* spp. In the current research, the trials shown under pot, field, and in vitro conditions are also mentioned in Figure 2.

*Biodiversity of Salt-Tolerant Plant Growth-Promoting Bacteria [PGPB]*

Soil and plant microbiomes possess a diversity of microbes belonging to diverse bacteria, archaea, and fungi [32]. Similarly, salinity-prevalent soil and crops facilitate halotolerant microbes. Among microbes, halotolerant plant growth-promoting bacteria demonstrate their inherent metabolic capability to tolerate salinity stress and promote plant growth through different mechanisms. Several researchers have reported on the diversity, tolerance mechanisms, and plant growth-promoting-predominant halotolerant bacterial genera, such as *Agrobacterium*, *Arhrobacter*, *Ochromobacter*, *Azospirillum*, *Alcaligenes*, *Bacillus*, *Enterobacter*, *Burkholderia*, *Microbacterium*, *Klebsiella*, *Streptomyces*, *Pseudomonas*, *Rhizobium*, and *Pantoea*, which have been reported to alleviate salt stress in crops [33–35]. In a study by Phour et al. [36], the inoculation of halotolerant *Pseudomonas argentinensis* HMM57 and *Pseudomonas azotoformans* JMM15 in *Brassica juncea* resulted in a significant increase in the shoot, root, and plant dry weights [36].

Zhang et al. [37] explored the genetic diversity of salt-tolerant PGPB (150 g/L Na$^+$Cl$^-$) isolated from a paddy rhizosphere in Taoyuan and China. The phylogenetic analysis showed 74 salt-tolerating isolates belonging to the orders, *Rhizobiales* (1%), *Oceanospirillales* (4%), *Actinomycetales* (22%), and *Bacillales* (72%) that further effectively enhanced the growth and productivity of rice under salinity stress. Aslam et al. [38] revealed that a halotolerant *Staphylococcus jettensis* F-11 significantly increased [threefold] the plant biomass (*Zea mays*), in dry weight, under 200-mM salinity stress. A current study demonstrates that the occurrence of salinity-tolerant *Pseudomonas putida* and *Novosphingobium* sp. reduce the negative effects of salt stress in citrus plants by lowering the levels of salicylic acid (SA) and abscisic acid (ABA). IAA's growing stock in the leaf inhibits photosystem II's processing (Fv/Fm) and prevents root congregation proline and chloride during salt stress [39]. Several researchers have documented the diversity, salt tolerance, and plant growth-promoting activity of coastal-soil-isolated PGPB, and their application as inoculates in pot and field experiments [40]. Amaresan et al. [41] deciphered the diversity of 121 bacterial strains isolated from Tsunami-affected regions in the Andaman and Nicobar Islands of India. The selected 23 isolates displayed salinity tolerance (10% Na$^+$Cl$^-$), with PGP characteristics, and belonged to the genera, *Lysinibacillus* spp., *Bacillus* spp., *Microbacterium* spp., *Alcaligenes faecalis*, and *Enterobacter* spp., and they are applied as bioinoculants for salinity-mitigation approaches in different crops. Several examples of halotolerant plant-beneficial microbes, their diversity, tolerance mechanisms, and plant growth-promoting responses are documented in Table 1. The role of salt-tolerant PGPB in the mitigation of salinity stress and the enhancement of crop productivity is well-studied. However, the interactions, colonization, and mechanisms between these microbes and plants, and the up-regulation of genes during salinity stress, need to be explored.

**Table 1.** PGPB with their salt tolerance values.

| Crop | Strain | Na$^+$Cl$^-$ Tolerance(mM) | Condition | Effect | Country | Reference |
|---|---|---|---|---|---|---|
| Wheat (*Triticumaestivum*) | *Azotobacter chroococcumAZ6* | 200 | Pot | Proline and amino acid | Algérie | [42] |
| | *Bacillus Subtilis* | 200 | Field | Improvement in plant growth | Russia | [43] |
| | *Enterobacter cloacaeZNP-3* | 150–200 | Pot | Enlargement of agronomic traits and chlorophyll content | India | [44] |
| | *Enterobacter cloacae SBP-8* | 200 | Pot | Antioxidant activity | India | [45] |
| | *P. fluorescens* | 18–36 | Invitro | Enhancement of agronomic traits | Iran | [46] |
| | *Pseudomonas sp.* and *Bacillus* sp. | Soil salinity | Pot | Plant growth promotion | India | [47] |
| Rice (*Oryza sativa*) | *Bacillus* and *Pseudomonas* spp. | Saline soil | Field | Phytostimulation | India | [48] |
| | *Bacillus amyloliquefaciensRWL-1* | 120 | Pot | Increase in essential amino acids | Korea | [49] |
| | *Pseudomonas aeruginosa [PRR1 and PHL3]* and *Lysinibacillus* sp. [BPC2] | 100 mM | Pot | Germination percentage and root length | India | [50] |
| | *Curtobacterium albidumSRV4* | 100–300 | Pot | Improve photosynthetsis, osmolytes, and antioxidative enzymes | India | [51] |
| | *Enterobactersp. P23* | 0 to 200 | Invitro | Promote rice-seedling growth | India | [52] |
| | *Enterobactersp. PR14* | 150–900 | Invitro | Growth promotion | India | [53] |
| Maize (*Zea mays*) | *Pseudomonas* | 150 | Pot | Improve POD activity, proline, and soil moisture | Pakistan | [54] |
| | *Enterobactercloacae [KP226569]* | 100–200 | In vitro | Increased growth parameters | India | [55] |
| | *Azotobacternigricans [KP966496]* | 100–200 | In vitro | Enlargement, growth parameters, and germination | India | [56] |
| | *Bacillus subtilis* | 0–200 | Plastic pots | Increased the relative water content in leaves | Brazil | [57] |
| | *Rhizobium tropici* and *Azospirillum brasilense* | 170 | Leonard jars | Antioxidant enzymes | Brazil | [58] |
| | *Azospirillum lipoferum* and *Azotobacter chroococcum* | 100 | Pot | Significantly enhanced growth parameters and pigments | S.Arabia | [59] |
| Millets | *E. cloacae* [KP226575] | 100–200 | Greenhouse | Increase in seed germination andenhanced root and shoot elongation | India | [60] |

**Table 1.** *Cont.*

| Crop | Strain | Na$^+$Cl$^-$ Tolerance(mM) | Condition | Effect | Country | Reference |
|---|---|---|---|---|---|---|
| Sorghum (*Sorghum bicolor*) | *Azotobacter salinestris* NBRC 102,611 | 50 | Pot | Improved plant growth, carbohydrate, proline, and macroelements | Egypt | [61] |
| Mung bean (*Vigna radiata*) | *Enterobacter cloacae* strain KBPD | 80 | Pot | Increase growth parameters, fresh and dry weights | India | [62] |
| Tomato (*Solanumly copersicum*) | *Azotobacter chroococcum* | 100 | Jar | Employed as bioeffectors | China | [63] |
| | *Azotobacter chroococcum* 76A | 100 | Pot | Promoted plant growth | Italy | [64] |
| | *Pseudomonas* spp. OFT2 and OFT5 | 75 | pot | Promoted shoot uptake of other macronutrients and micronutrients | Japan | [65] |
| Pea (*Pisum sativum*) | *Bacillus subtilis* RhStr_71, *Bacillus safensis* RhStr_223, and *Bacillus cereus* RhStr_JH5 | 10 | Pot | Enhanced the antioxidant enzymes | India | [66] |
| | *Enterobacter ludwigii Acinetobacter bereziniae*, and *Alcaligenes faecalis* | 150 | Field trials | Seedling enlargement and yield | India | [67] |
| Chickpea (*Cicerarietinum*) | *Bacillus subtilis* [BERA 71] | 200 | Plastic pots | Increased the synthesis of photosynthetic pigments and plant biomass | Saudi Arabia | [68] |
| | *Rhizobium* sp. | 150 | Pot | Influenced photosynthesis and improved yield attributes | India | [69] |
| Capsicum (*Capsicum annuum* L.) | *Bacillus* WU-9, WU-5, and WU-13 | 60 | Pot | Growth parameters and freshand dry weights | china | [70] |
| Red Pepper (*Capsicum annuum*) | *Pseudomonas frederiksbergensis* OB139 and *vancouverensisOB155* | 150 | Plastic pots | Increased content of photosynthetic pigments | Republic of Korea | [71] |
| Okra (*Abelmoschuse sculentus*) | *Enterobacter sp.* UPMR18 | 75 | Plastic pots | Increased antioxidant enzyme activities | Malaysia | [72] |
| Shalgam (*Brassica rapa*) | *Pseudomonas stutzeri* ISE12 | 200 | Pot | Antioxidant defense system | Poland | [73] |
| Soybean (*Glycine max*) | *Bacillus subtilis* and *P.pseudoalcaligenes* | 100 | Hydroponics | Enhancement of growth parameters | India | [74] |
| Soybean | ALT29 and ALT43 | 80–2400 | Pot | Improvement in agronomic parameters, fresh and chlorophyll content | Korea | [75] |

**Table 1.** *Cont.*

| Crop | Strain | Na⁺Cl⁻ Tolerance(mM) | Condition | Effect | Country | Reference |
|---|---|---|---|---|---|---|
| Canola (*Brassica napus*) | *Enterobacter cloacae* [HSNJ4] | 50–100 | Pot | Enhancement of IAA and ethylene content | China | [76] |
| Canola (*Brassica napus*) | *Rhizobium* sp. | 90 | GH | The increased dry weight of the root | Iran | [77] |
| Faba bean (*Viciafaba*) | *Rhizobium leguminosarum* | 0–120 | Marrakech-Haouz region | Increased plant biomass, nodule number, and nitrogen content | Morocco | [78] |
| Snap Bean (*Phaseolus vulgaris*) | *Rhizobium*–arbuscularmycorrhizal fungi [AMF] | 6–200 | Organic garden-shade house | Increased yield | Florida | [79] |
| Pegion Pea (*Cajanus cajan*) | *Funneliformis mosseae* and *Rhizophagus irregularis* | 0–100 | Pot | Symbiosis | India | [80] |
| Hopbush Shrub (*Dodonae aviscosa* L.) | *Azospirillumlipoferum* and *Azotobacterchroococcum* | 150–200 | Pot | Improved seedling growth parameters | Iran | [81] |
| *Artemisia annuaL.* | *Piriformospora indica* [Pi] and *Azotobacter chroococcum* [Az] | 200 | Pot | Enhanced activities of antioxidant enzymes | India | [82] |
| Sunflower (*Helianthus*) | *Pseudomonas citronellolis SLP6* | 80 | pot | Increase in antioxidant enzymes and chlorophyll content | India | [83] |
| *Arabidopsis thaliana* | *Pseudomonas putida* | 200 | Pot | Auxin Activity RAR shows tryptophan trust for growth | India | [84] |
| Alfalfa (*Medicago sativa* L.) | *S. meliloti* ARh29, *K. cowanii* A37, *Klebsiella* sp. A36, and rhizobial strain | 1200 | Greenhouse and invitro | Plant growth | Iran | [85] |
| Alfalfa (*Medicago sativa* L.) | *B. subtilis* NRCB002, NRCB002, NRCB003 [NRCB003] | 130 | Greenhouse | Significantly increased dry weight | China | [86] |
| *Medicago sativa* | *Klebsiella* sp., *Alcaligenes* sp., *Pseudomonas cedrina*, and *Pseudomonas putida* | 20–60 | Pot | Plant development, chlorophyll content, and root AM colonization | Morocco | [87] |
| | *Pseudomonas aeruginosa* | 170 | Field experiments | Significant increase in shoot length | Pakistan | [88] |
| Talh Tree *Acacia gerrardiiBenth.* | AMF and *B. subtilis* | 250 | Pot | Increase in phenols, fiber content, and total lipids | Riyadh, KSA | [89] |

## 4. Mechanisms of Halotolerant PGPB-Mediated Salinity-Stress Tolerance

Numerous systems have been accounted for by Halotolerant PGPB for the exercises of salinity amelioration [18]. Thehalotolerant PGPB exploits numerous systems (Figure 3) that indirectly or directly participate in the amendment of salinity [90]. Halotolerant PGPB produces numerous plant growth regulators, such as cytokinins, auxins [IAA], gibberellins. PGPB produce ACC deaminase [90], exopolysaccharides [91], and siderophores [92–94]. They also mobilize nutrients, such as phosphorous solubilization [95], nitrogen-fixing [96,97], and osmolytes (trehalose, proline, and glycine betaines), activate the antioxidative enzymes of plants, and regulate plant defense systems under salt stress [98] (Figure 3).

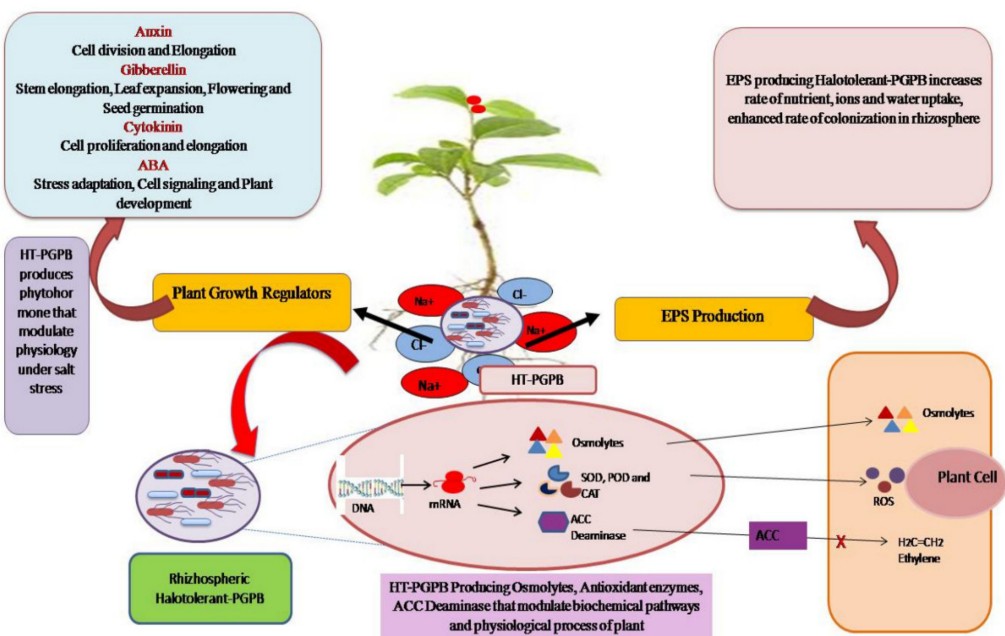

**Figure 3.** The alleviation of salt stress in plants mediated by halotolerant PGPB.

### 4.1. Phytohormone Production

Phytohormones secreted by halotolerant PGPB play a vital role in regulating plant metabolic pathways and physiological processes under salinity conditions. There are reports where halotolerant PGPB is known to produce more than one type of phytohormone. Gibberellin-positive bacterial strains, such as *Bacillus. licheniformis*, *Azospirillium* spp., and *B. pumilus* havealso been identified by recent studies [99,100]. Patel and Saraf [101] report that bacterial strains of *Stenotropho monasmaltophilia*, *P. stutzeri*, and *P. putida*, isolated from the *Coleus* rhizosphere, produced gibberellic activity, IAA, and CK in salt environments. In recent times, Tewari et al. [102] documented the involvement of SA in improving salt stress and promoting the enlargement of sunflower in alkaline soils.

#### 4.1.1. Auxin [Indole Acetic Acid, IAA]

The biosynthesis of IAA occurs via multiple pathways in halotolerant PGPB that convert precursor tryptophan into IAA, which is used by the plant and which stimulates cell growth and proliferation under salinity conditions [103]. IAA produced by halotolerant PGPB is the most ordinary and broadly studied. Phytohormones inside salt stress mitigate plant–microbe interactions [18]. There are several well-known IAA-producing halotolerant PGPB under salt stress: *Stenotrophomonas* spp., *Pseudomonas* spp., *Rahnella* spp., and *Bacillus* spp. [104,105]. Inoculation of the halotolerant PGPB has also been shown to protect plants from ion toxicity, improve mineral absorption, and increase growth parameters in saline environments [106]. Several recent studies have established that PGPB produces both ACC deaminase and IAA under salt stress to successfully protect plant growth from salinity

stress. The accumulation of IAA in plant tissues stimulates transcriptional factors related to ACC synthase genes, which enhances the concentration of ACC, and, in response, the production of ethylene increases [107]. *Pantoea dispersa* PSB3 is an indigenous bacterium in chickpea (*Cicer arietinum*) and produces ACC deaminase and IAA. It significantly increases the seed number, the plant biomass, the seed weight, and the salt in the pods of affected plants [108]. *Bacillus tequilensis*UPMRB9 was the best IAA producer for enhancing the root surface and for modifying the root morphology to obtain more nutrients and water from the soil [109]. In addition to EPS production, this breed's exceptional IAA production is vigorous for adventitious roots and lateral root growth, resulting in increased nutrient availability and water absorption. Banerjee et al. [110] conclude that the inoculation of the IAA-producing isolates of rice plants results in significant root- and seedling-length enlargements. Recently, Kang et al. [107] demonstrated that the IAA-producing halotolerant PGPB, *Leclercia adecarboxylata*, has been characterized for sugar activity and chlorophyll fluorescence development in tomato.

### 4.1.2. Cytokinin

Apart from IAA, cytokinins are compulsory phytohormones for the cell-cycle succession and for mitigating the influences of salinity stress in plants [99]. The cytokinins are involved in various physiological functions of plants, such as cellular proliferation and differentiation, chlorophyll biosynthesis, and chloroplast biogenesis [111]. Inoculating cytokinin-producing bacteria enhances the shoot and fruit formation, as well as the plant resistance to abiotic stress. Several researchers have reported salt-tolerant PGPB that mitigate salinity through cytokinins, including *Halomonas Arthrobacter*, *Pseudomonas* sp., *Bacillus* sp., and *Azospirillum* species [112–114]. Furthermore, our findings were strengthened by the results that show that increased resistance to the abiotic stresses of *Platycladus orientalis* occur with the cytokinin-producing *Bacillus subtilis*. [115]. ABA is produced by numerous strains of ST-PGPR, comprising *Pseudomonas fluorescens*, *Bacillus licheniformis*, *B.megaterium*, *Proteus mirabilis*, and *Achromobacter xylosoxidans*, which disrupts the activity of cytokinin in root-to-shoot signaling [116]. The up-regulation of cytokinin synthesis genes promoted an increased production of cytokinins in tomato plants, reflecting the organic volatiles excreted by the PGPB *Bacillus subtilis*, SYST2, under salinity-stress conditions [113].

### 4.1.3. Antioxidant Activity

Several researchers report that halotolerant PGPB-inoculated plants combat the deleterious impacts of oxidative stress by producing antioxidative enzymes [25,117,118]. Several antioxidant mechanisms demonstrated by the halotolerant PGPB-involved enzymes, such as glutathione reductase, peroxidases nitrate, superoxide dismutase, and catalase, as well as secretory molecules, protect tissues from oxidative damage by suppressing and detoxifying ROS [66,119–131]. There are many PGPB strains, such as *Rhizobium* sp., *Azotobacter* sp., *Pseudomonas* sp., and *Serratia* sp., which can be used as inoculants for improving plant growth under salinity stress [75,122,132]. The upregulation of the antioxidant gene responses, in terms of antioxidant production and metabolite synthesis, are measured as the sensitive physiological markers of salt stresses [123,124]. El-Esawi et al. [125] demonstrate that *Azospirillum lipoferum* FK1-inoculated plants exhibited the advanced expression of antioxidant genes and, therefore, improvements in the nutrient uptake, phenol, flavonoid content, antioxidant enzymes, and non-enzymatic metabolize, enlargement, and the rising of chickpeas. In addition, an antioxidant enzyme involved in ROS scavenging activities and ascorbate peroxidase showed enhanced tolerance in tomato plants inoculated with halotolerant PGPB under salt stress [117]. Habib et al. [72] report the enhanced salt tolerance of okra (*Abelmoschus esculentus*) plants when inoculated with *Enterobacter* sp. UPMR18 increased antioxidant enzyme activity, and the ROS cascade gene expression was shown when inoculated with 75 mM of $Na^+Cl^-$. Several documented ST-PGPR, including *Pseudomonas pseudoalcaligenes* and *Enterobacter cloacae*, improved the levels of CAT and APXin Jatropha leaves in response to salinity stress and also increased

the biomass, encouraged the development of the root, potassium [K], and phosphorus, the uptake of nutrients, and the chlorophyll content in different tissues of the plant [15].

Polyamines [PAs] are another abundant antioxidant with low molecular weight aliphatic amines, ubiquitously present in plants/microbes, which regulate ROS homeostasis by scavenging free radicals by triggering antioxidant enzymes. The researchers report different polyamines, such as spermidine, putrescine, and spermine, associated with numerous developmental processes and stress responses in plants [126]. The antioxidant system also comprises non-enzymatic systems, such as salicylic acid and carotenoid production [127]. Similarly, Cappellari et al. [128] observed a decreased level of malondialdehyde (MDA, an indicator of lipid peroxidation and membrane damage) in peppermint seedlings under saline conditions when treated with microbial volatile organic compounds (mVOC)and reduced oxidative damage through an increased production of the antioxidant molecule, [DPPH-2,2-diphenyl−1-picrylhydrazyl]. Side by side, the findings observed by recent research suggest the significance of Halotolerant PGPR in plant stress tolerance through the modulation of biochemical processes and plant physiology, for example, osmolytes, stress-related genes, and enzymatic and non-enzymatic antioxidants. Abdelaal et al. [129] describe that salt stress adversely affected the relative water content, the chlorophyll content, and the fruit yields, whereas proline-concentration-electrolyte leakage, malondialdehyde [MDA], reactive oxygen species [ROS], and the activities of antioxidant enzymes increased in salt-stressed plants. Several studies have recognized the capability of microbes to moderate the effects of oxidative reuptake by increasing the osmolyte content and the activity of antioxidant enzymes [117,123,124,127,130–140].

### 4.1.4. Osmoprotectants

Osmoprotectants (compatible solutes with neutral charges) are low-molecular-weight compounds that help plants or microorganisms survive in adverse saline conditions. Usually, plants gatherorganic-natured compound osmolytes, including quaternary ammonium compounds, polyamines, glycine, betaine, proline, and other amino acids, in response to salinity stress [131]. Halotolerant PGPB employ a similar mechanism counter to osmotic stress, the collective phenomenon of alkaline soils [132]. Although subjected to salt stress, halotolerant bacteria may upsurge their $K^+$ions, the accretion of osmolytes, and their cytoplasmic content to help the persistent stress response and to avoid water loss [133]. Recently, Kushwaha et al. [134] revealed the role of osmoprotectants in (SBS 10) *Halomonas* sp. and showed that low $Na^+Cl^-$ concentrations accelerate the betaine level that overturns the de novo production of ectoine; however, at high $Na^+Cl^-$ concentrations, the ectoine concentration increases brusquely as compared to the betaine.

Furthermore, they determined that the accumulation of ectoine is regulated by transcription elements and that it enhances the expression of transcription factors during salinity stress. In other studies, it was perceived that *Azospirillum* spp. produce proline, betaine, glycine, trehalose, and support plants in order to endure osmotic tension. A halotolerant PGPB strain of *Bacillus* sp. was documented to produce proline and soluble sugars, and significantly enhanced the growth of maize under salinity and drought conditions [18]. Several researchers report the significance of the trehalose as an osmoprotectant and of producing halotolerant PGPB that comprises the genes for trehalose biosynthetic pathways under salt stress [135–137].

### 4.1.5. Exopolysaccharide (EPS) Production

Some rhizosphere bacteria produce exopolysaccharides (EPSs) or surface polysaccharides, which is a common feature. However, the composition and quantity of EPS might differ in diverse halotolerant PGPB cultures; ample EPS is secreted in hostile conditions, which work as a physical obstruction near the root and protect plant growth in high-saline conditions [16]. Singh and Jha [44] report the enhanced uptake of ions in plants under saline conditions when inoculated with EPS-producing halotolerant PGPB. The ACC-deaminase and EPS-producing halotolerant PGPB, *Mesorhizobium ciceris*, enhanced chickpea growth,

stabilized the soil texture, and enhanced colonization under salinity. In the context of yield improvement, the role of EPS-producing halotolerant PGPB is significant as they are being used as the priming agents of seeds and to help enhance germination [98]. Atouei et al. [138] found that the inoculation of wheat with EPS-producing and salt-tolerating *Marinobacter lipolyticus* SM19 and *B. subtilis* decreases the adverse effects of abiotic stresses. In recent times, Chu et al. [139] observe that the EPS-producing ST-PGPR, *Pseudomonas* PS01, improved *Arabidopsis thaliana* growth and the regulation of genes under salinity stress. They found that a LOX2 gene was up-regulated and that it participated in the jasmonic acid [JA] pathway and enhanced the accumulation of JA plants under abiotic stress, whereas bacterial EPS provides complimentary profit for tolerance below salt stress.

4.1.6. 1-Aminocyclopropane-1-Carboxylic Deaminase [ACCD]

Halotolerant plant growth-promoting bacteria secrete enzymes to reduce salinity stress, including reducing ethylene levels via hydrolyzing 1-aminocyclopropane-1-carboxylic acid [ACC] by the enzyme, ACC deaminase. ACC is the immediate precursor of the hormone ethylene in plants. It is widely reported that PGPB possesses ACC deaminase enzymes that can degrade ACC to ammonia and $\alpha$-ketobutyrate, thus reducing the levels of ethylene inside the plants [37,100,140–142]. Bacterial ACCD activity, or the induced systemic resistance in plants are both factors that would have led to reduced ethylene biosynthesis [142]. The increased SOD activity of wheat tissues [both root and seedling] in plants planted with ACC-producing rhizobacteria suggests better antioxidant performances, stimulated by ACC-producing rhizobacteria, thereby reducing the harmful effects caused by ROS. Several studies have shown that bacterial consortium inoculation with high ACCD activity and auxin production can modify and improve the antioxidant system in wheat seedlings. ACCD-producing bacteria are classified as halotolerant, with the ability to grow in the range of 2–11% $Na^+Cl^-$ [44,63]. Wheat seedlings planted with two selected ACCD-producing *Klebsiella* sp., 8LJA and 27IJA, significantly ($P \leq 0.05$) enhanced the biomass (45–62%) content and the SOD activity in roots (18–35%), under normal as well as under salt conditions [143]. The seed bacterization by ACC02 and ACC06 reveals that the treatment of plants with bacterial consortia significantly declined the salinity. stress through ACC deaminase activity (approximately 60%) and stimulated the ethylene levels [144]. Several reports have recommended that the function of ACC deaminase producers manifest salt stress in plants by reducing ethylene emissions and by providing plant growth promotion and stress tolerance in stressed plants [46,145].

**5. Role of Halotolerant PGPB in the Enhancement of Crop Production under Salinity**

It is now widely accepted that halotolerant PGPB has the inherent potential to cope with high-saline conditions and to facilitate plant growth through different direct/indirect beneficial mechanisms [50]. Halotolerant PGPB have been reported as bioinoculants/bioformulations, biostimulants, and biofertilizers that enhance the crop yields and facilitate the survival of plants, even under high-temperature concentrations of salt [146,147] (Table 2).

They are present in the soil and directly promote the plant growth of many types of cereals and other important crops [29,32,147]. It has been realized that significant crops, for example, cereal crops (maize, barley, oats, sorghum, rice, wheat, and millet), legumes, vegetables, and oil-yielding crops, are affected by salinity, but that the inoculation with the halotolerant PGPB not only helped in maintaining the crops, but also increased the yields [44]. The role of Halotolerant PGPB in increasing the productivity of several crops is discussed in this section.

**Table 2.** Role of Halotolerant PGPB in the enhancement of crop production.

| Crop | Halotolerant PGPB | Tolerance Mechanism | Reference |
|------|------|------|------|
| | Cereals | | |
| Rice | *Thalassobacillus denorans* and *Oceanobacillus kapialis* | Increase in germination % | [148] |
| | strains *S6* and *S7* | Yield increased | [149] |
| | *Enterobacterudwigii* [AF134] and *P. Putida* [AF137] | Increased root and shoot biomass | [150] |
| | *Enterobacter cloacae* [KP226569] | Increase in germination percentage, root, and shoot | [151] |
| | *Azotobacter* sp. [KH-2] | Enlargement of plant height, yield, and biomass | [152] |
| | *Alcaligenes* sp. [AF7] | Increasingly support the vegetative growth parameters of crop | [153] |
| | *Exiguobacterium* sp., *Stenotrophomonas* sp, *Enterobacter* sp., *Microbacterium* sp., and *Achromobacter* sp. | Significantly improve proline, total chlorophyll, and total phenol | [154] |
| | *Glutamicibacter* sp. [YD01] | Enhanced stress-responsive gene expression and photosynthetic | [155] |
| | *Bacillus tequilensis* [UPMRB9], *Providencia stuartii* [UPMRG1], and *B. aryabhattai* [UPMRE6] | Improved chlorophyll contentand reduced electrolyte leakage | [156] |
| Wheat | *P. fluorescens*, *B. pumilus*, *Exiguobacterium aurantiacum* | Enhancement of agronomic traits | [157] |
| | *Klebsiella* sp. | Increase in total protein content and proline | [158] |
| | *Azospirillum brasilense* 65B [Abr65B] | Enhanced enlargement and nutrient access | [159] |
| | *Erwinia* sp. | Enhanced growth and yield parameters | [160] |
| | *Bacillus megaterium*, *P fluorescens* and *B. subtilis* | Increased grain yield | [161] |
| | *Azospirillum* sp., *Azoarcus* sp., and *Azorhizobium* sp. | Increase in biomass and the activity of enzymes | [162] |
| | *Azotobacter* sp. [Az1-Az6] | Improved agronomic characteristics | [163] |
| Maize | *Staphylococcus sciuri* | Significantly increased growth parameters and biochemical characteristics | [164] |
| | *Pseudomonas* sp. P8, *Peribacillus* sp. P10, *Streptomyces* sp. X52 | promoted the growth | [165] |
| | *Bacillus aquimaris* DY-3 | Increased chlorophyll content antioxidant enzymes and osmotic regulation | [166] |
| | *Bacillus* sp. SR-2-1/1 | increased contents of chlorophyll, total phenolics and proline | [167] |
| | *Serratialiquefaciens* [KM4] | Antioxidant enzymes, decrease in ABA biosynthesis, and nutrient uptake | [168] |
| | *Bacillus* sp. [HL3RS14] | Increased dry weights of roots and shoots | [169] |
| | *B.safensis* [NBRI 12 M] | Increased amounts of chlorophyll, proline, and soluble sugar | [170] |
| | *Bacillus* sp. [NBRI YN4.4] | Improved biochemical traits and soil enzymes | [171] |
| Oat | *Klebsiella* sp. [IG 3] | Enhanced plant growth | [172] |

**Table 2.** *Cont.*

| Crop | Halotolerant PGPB | Tolerance Mechanism | Reference |
|---|---|---|---|
| | Oil-yielding grains | | |
| Soybean | *Bradyrhizobium* sp. | Increased grain yield | [173] |
| | *P. fluorescens* [LBUM677] | Improved plant biomass, lipid composition, and oil content | [174] |
| | *Bradyrhizobium* sp and *L. adecarcoxylata* | Enhanced plant growth and productivity | [175] |
| | *B. velezensis* [S141], and *B.diazoefficiens* [USDA110] | Improved nodulation and $N_2$-fixing efficiency | [176,177] |
| Sunflower | *Bradyrhizobium* sp. | nodY/K and nifH gene expression for salinity tolerance | [178] |
| Groundnut | *Klebsiella*, *Pseudomonas*, *Agrobacterium*, and *Ochrobactrum* | Enhanced PGP traits | [179] |
| Cotton seed | *B. amyloliquefaciens*, *Curtobacterium oceanosedimentum*, and *Pseudomonas oryzihabitans* | Seed germination | [180] |
| | Legumes | | |
| *Chickpea* | *Bradyrhizobium* and *Actinomadura*, *Paenibacillusgraminis* | Induced increased antioxidant enzymes, and | [181] |
| *Peanut (Arachish ypogaea L.)* | *Serratia* sp. S119; *Acinetobacter* sp. L176; | Increased growth parameters and P content | [182] |
| | *Planomicrobium* sp. MSSA-10 | Increased growth | [183] |
| *Black gram* | *P. fluorescens* | Improved photosynthetic content | [184] |
| *Faba bean* | *Pseudomonas anguilliseptica* [SAW 24] | EPS production and biofilm production | [185] |
| | Vegetables | | |
| Tomato | *Bacillus pumilus* | Significantly improved the fresh shoot weight and dry weight | [186] |
| | *B. thuringiensis* | Seed germination and shoot elongation | [187] |
| | *Pseudomonas jessenii* and *Pseudomonas synxantha* | Significantly enhanced plant growth | [188] |
| | *Pseudomonas* sp. KP966497 [PR21] | Seed germination and shoot elongation | [189] |
| | *Pseudomonas* sp. UW4 | Enhanced plant growth | [190] |
| Chinese cabbage, Lettuce, Radish | *Lactobacillus* sp. and *P. putida*, *A. chroococcum* | Increased the plumule and radicle lengths of germinated seeds | [191] |
| Capsicum | *B. fortis* strain [SSB21] | Reduced phytohormones | [184] |
| Potato (*Solanum-tuberosumL.*) | *Bacillus* strains | Enhancement of plant growth | [192] |

## 6. Molecular Understanding of Halotolerant PGPB-Mediated Salt Tolerance in Plants

Different genes are affected by PGPB to confer salt tolerance in plants. The efficient colonization of *Enterobacter* sp. EJ01 in tomato and *Arabidopsis* plants alleviate salt stress by stimulating the expression of responsive genes, including the RD29A, DREB2b, and RAB18 genes related to the ABA-dependent pathways. There is scientific evidence that Halotolerant *Pantoeaag glomerans* defend tropical corn plants from salt stress by upregulating the expression of aquaporin genes (*zmPIP2-1*, *zmPIP1-1*, and *zmPIP2-5*). Similarly, the induction of salt tolerance and the stimulation of a set of genes accountable for salinity tolerance in wheat by the halotolerant *Dietziana tronolimnaea* STR1 strain have been reported by Bharti et al. [25]. The halotolerant PGPB-inoculated plants always have higher SOS gene expression under salt conditions than uninoculated salt-stressed plants, proposing their involvement in salt tolerance mechanisms. *SOS4* gene expression is associated with the auxin and ethylene mechanisms, as well as root hair development in *Arabidopsis*. The improved SOS4 gene up-regulation in PGPR-inoculated wheat roots can thus be correlated with prolific growth in the root length and the overall plant growth in salt-stressed plants [25]. The dominant pathways, such as ion transporters, ABA-signaling, osmoprotectants, the SOS pathway, and antioxidant machinery usually protect the plant from oxidative damage caused by high salt concentrations. These are accessed by the up-regulation of salt-tolerant genes, such as transcriptional factors (TaABARE, TaWRKY, and TaOPR1), ion transporters (TaHKT1, TaNHX1, and TaHAK), and SOS-pathway-related genes (SOS1and SOS4) [24]. The activation of the WRKY gene has been reported in response to salinity stress in soybean [193], and arabidopsis [194]. Wheat plants inoculated with the PGPR strains, *B. subtilis* and *Arthrobacter protoformia*, under salinity-stress conditions produced IAA content and there was an observed upregulation of the TACTR1 and TADRE2 genes to reduce salinity stress [195]. The inoculation of the halotolerant PGPB strain, IG3, in oat under salinity conditions increased the auxin concentration in plants under the same condition. It was noted that an increase in the concentration of auxin in tomato plants helped to improve the osmotic stress generated by salinity [196]. de la Torre-González et. al. [196] decoded the expression of the cytokinin synthesis gene, SlCKX1, and promoted high cytokinin production when tomato plants were exposed to the halotolerant PGPB *Bacillus subtilis* SYST2 strain exuding organic volatiles. The knockdown of cytokinin oxidase, OsCKX2, in rice, using the RNAi-based approach, resulted in a significant increase in cytokinins under salt-stress conditions. Cytokinin oxidase, OsCKX2, knockdown plants displayed enhanced vegetative growth, photosynthetic efficiency, and relative water content compared to wild types under salinity stress [197]. Asaf et al. [198] decoded the whole genome sequence of the salt-tolerant endophytic *Sphingomonas* sp., LK11, containing many genes encoding glycine betaine from choline by encoding the betTcholine transporter, the betacholine dehydrogenase, and the betaine aldehyde dehydrogenase, and further validating the genetic role of the LK11 gene in resisting salinity stress and promoting plant growth. Similarly, Chen et al. [199] report the expression of an upregulated $Na^+/H^+$ antiporter, [NHX], and $H^+$-PPase genes under saline conditions in *Bacillus amyloliquefaciens* SQR9-inoculated maize, and enabled $Na^+$ confiscation into the vacuoles of maize shoots. The recirculating of $Na^+$ from shoot to root via an elevated expression of the high-affinity $K^+$ transporter 1 [HKT1] gene was observed in the treated maize plants. Forni et al. [136] demonstrated the potential role of the salt-tolerant strain, PS01 *Pseudomonas*, causing EPS in regulating genes connected to stress tolerance in *Arabidopsis thaliana* [194]. Upregulated genes related to transcription factor [WRKY] modulate osmotic balance, scavenge ROS, and reduce salinity and drought stress by triggering stress-related genes. The study also proved that the inoculation of wheat with *A. nitroguajacolicus* had a higher WRKY28 gene expression and showed significant crop growth under salinity stress. Similarly, Kazerooni et al. [200] reveal that halotolerant PGPR-inoculated pepper plants showed the upregulated gene expression of *XTH* (Xyloglucan Endotrans glucosylase/Hydrolase) and the decreased expression of *BI-1*, *PTI1*, and *WRKY2*, and the binding immunoglobulin protein (BiP) genes (*CaBiP1*, *CaBiP2*, and *CaBiP3*), under salinity and

drought conditions and enhanced plant growth. In contrast, PGPR-uninoculated pepper seedlings showed decreased cell-wall extensibility and retardation in plant growth. In another study, halotolerant PGPR-treated stressed plants displayed higher XTH (XTH1 and XTH3) gene expression to tolerate salt and drought stress in tobacco and pepper plants and revealed plant heights and leaf lengths/widths. In this order, Liu et al. [201] demonstrate the expression of salinity-induced genes related to the Aux, CK, and GA signaling pathways, as well as the WRKY and MYB transcription factors (TFs) in *Paenibacillus polymyxa* YC0136-inoculated tobacco plants (*Nicotiana tabacum* L.) and promote plant tolerance in stress conditions. Akram et al. [202] demonstrate the multiple mechanisms in the *Bacillus megaterium* strain, A12, for the amelioration of salinity stress in tomato plants. In this order, several genes of halotolerant PGPB, their gene products, tolerance mechanisms, and related crops are documented in Table 3.

**Table 3.** Genes for salt-stress tolerance in different PGPB.

| Halotolerant PGPB | Crops | Gene | Gene Product/Protein/ Enzyme | Significance | Reference |
|---|---|---|---|---|---|
| *Bacillus megaterium* | Tomato | Expression of MT2 and GR1 | Metallothionein Glutathione reductase enzyme synthesis | Antioxidant enzyme production, tolerance to salinity | [203] |
| *Bacillus megaterium* strain A12 [BMA12] | Tomato | *PsbA* and *PBGD* | | Increased biosynthesis of chlorophyll | [203] |
| *Pseudomonas putida PS01* | *A. thaliana* | Upregulation of *LOX2* | Jasmonic acid synthesis pathway, SOS pathway, abscisic acid [ABA] production | ROS activity, abscisic acid | [139] |
| *Arthrobacter nitroguajacolicus* | Wheat | Upregulation of AA0618700, AA0359620, APX, and GPX Expression of AA0410390, AA1982260, AA0412840, and AA1872340 | SOS pathway and antioxidant enzyme | Plant-cell-wall biosynthesis, ROS activityosmotic balance | [117] |
| ALT29 and ALT43 | Soybean | [GmFLD19 and GmNARK] | Antioxidants [GSH, LPO, TPP, PPO, and POD], ion uptake [ Na and K] | Significantly increased growth parameters and biochemical traits | [75] |
| *Bacillus amyloliquefaciens* | Pepper | Upregulation of *XTH* [Xyloglucan Endotrans glucosylase/Hydrolase] | High biochemical traitsand antioxidant activities | Enhanced plant growth and tolerance for salinity and drought conditions | [201] |

## 7. Conclusions and Future Prospects

Salinity stress causes high costs in agriculture production, and its destructive effects limit plant growth and reproduction, resulting in reduced food production. Plants have the inherent ability to respond to specific types of stress, but the application of PGPB to stimulate crop growth, through direct and indirect mechanisms, under salt-induced stress conditions, is an emerging strategy for enhancing global food production. Inoculated plants with PGPR make the plant more resistant to salinity, and PGPB strains boost the growth parameters and enhance plant productivity under salinity stress. PGPB play a vital role in enhancing plant enlargement and crop productivity, maintaining balanced nutrient cycling, controlling pesticide pollution, and producing secondary metabolize. The recent thirst of researchers is to develop a combination of formulations that will help stress tolerance and promote plant growth for sustainable agriculture. The use of advanced molecular techniques to search for halotolerant PGPB, to mediate the conservation of salt-affected soil, and for the exploration of biochemical mechanisms, genes, and signaling to neutralize salinity stress in crops can play a revolutionary role in applying halotolerant PGPR for growing crops under salinity stress.

**Author Contributions:** Conceptualization and writing—original draft: A.S. and R.Z.S. Formal analysis: S.R. Writing—review and editing: N.I., A.I.A.-T., H.A.E.E. and T.S. All authors have read and agreed to the published version of the manuscript.

**Funding:** This work was supported by RMC, Universiti Teknologi Malaysia, Malaysia through grant No. R.J130000.7344.4C336 and R.J130000.7609.4C468.

**Institutional Review Board Statement:** Not applicable.

**Informed Consent Statement:** Not applicable.

**Data Availability Statement:** All the data is available in the manuscript file.

**Acknowledgments:** Authors would like to thank RMC, Universiti Teknologi Malaysia, Malaysia for financial support through grant No. R.J130000.7344.4C336 and R.J130000.7609.4C468.

**Conflicts of Interest:** The authors declare that they have no conflicts of interest.

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
