# Peer review of "Halotolerant Rhizobacteria for Salinity-Stress Mitigation: Diversity, Mechanisms and Molecular Approaches"

_sustainability, doi:10.3390/su14010490_

Round 1

Reviewer 1 Report

The ms provides valuable information on use of halotolerant rhizobacteria for salinity stress mitigation. Therefore, these studies are useful for budding researchers of this field. However, please pay attention to the following points:

  • Abstract, line 3: It seems more appropriate to use “in many countries of the world” instead of
  • Abstract: Consider addition of future application of halotolerant Rhizobacteria.
  • The latter half portion in first para of Section 3.1 should be revised. The paragraphs may be reframed.
  • Table 1: ‘NaCl value’ should be replaced with ‘NaCl tolerance’ in the column head of 3rd  column. 
  • Consider revision of last sentence in the Conclusion section. 
  • Reduce number of references up to 200 and replace the old references with latest ones. 
  • Figures must be cross checked for standard resolution and quality of images, particularly quality of text in the images.

Reviewer 1 Report

 The ms provides valuable information on use of halotolerant rhizobacteria for salinity stress mitigation. Therefore, these studies are useful for budding researchers of this field. However, please pay attention to the following points:

 Abstract, line 3: It seems more appropriate to use “in many countries of the world” instead of

Authors response: Agreed and corrected

  • Abstract: Consider addition of future application of halotolerant Rhizobacteria.

Authors response: Added

 The latter half portion in first para of Section 3.1 should be revised. The paragraphs may be reframed.

Authors response: Reframed

  • Table 1: ‘NaCl value’ should be replaced with ‘NaCl tolerance’ in the column head of 3rd

Authors response: Corrected

  • Consider revision of last sentence in the Conclusion section. 

Authors response: Rephrased

  • Reduce the number of references up to 200 and replace the old references with the latest ones. 

Authors response: Agreed and the references of 2016 onwards are retained. Now there are total 206 references

  • Figures must be cross-checked for standard resolution and quality of images, particularly the quality of text in the images.

Authors response: Quality of figures and text in the figures is now improved

Reviewer 2 Report

This study provides valuable background information on Halotolerant Rhizobacteria for Salinity Stress Mitigation: Diversity, Mechanism and Molecular Approaches. Therefore, this study are important for researchers in this field. However, please pay attention to the following notes:

Abstract:

In line 3: Replace “in many countries of the world” instead of worldwide.

Introduction:

  • Reference No. 8: Using the latest version of the FAO Main Report.
  • In Main Title 1: Adding a paragraph on the technique of gene transfer from plants known to be highly salt tolerant to non-tolerant plants such as wheat.
  • In the headlines: from 3 to 7: Add an introduction, objective, and summary (if it is not available) at the end of each subheading and not just show reference studies on the topic.
  • Are the Figures in this study the work of the authors? Or have references been relied upon?
  • It is preferable to perform formatting in these Figures such as deleting blank spaces in text boxes

Note: Please see the notes in the attached file

Reviewer 2 Report

This study provides valuable background information on Halotolerant Rhizobacteria for Salinity Stress Mitigation: Diversity, Mechanism and Molecular Approaches. Therefore, this study are important for researchers in this field. However, please pay attention to the following notes:

 Abstract

  • In line 3: Replace “in many countries of the world” instead of worldwide.

Author response:  Replaced.  Line 23

Introduction

  • Reference No. 8: Using the latest version of the FAO Main Report.

Author response: added the latest report of FAO: Panagos, Panos; Borrelli, Pasquale; Robinson, David (2019). FAO calls for actions to reduce global soil erosion. Mitigation and Adaptation Strategies for Global Change, –. doi:10.1007/s11027-019-09892-3 

 In Main Title 1: Adding a paragraph on the technique of gene transfer from plants known to be highly salt tolerant to non-tolerant plants such as wheat.

Author response: Agreed to the suggestion. The information about technique of gene transfer from plants has been added. Line 94-96

  • In the headlines: from 3 to 7: Add an introduction, objective, and summary (if it is not available) at the end of each subheading and not just show reference studies on the topic.

Author response: Agreed and corrected

  • Are the Figures in this study the work of the authors? Or have references been relied upon?

Author response: All the figures have been prepared by the authors

  • It is preferable to perform formatting in these Figures such as deleting blank spaces in text boxes

Author response: Figures are formatted properly and all the blank spaces have been deleted.

  • Note: Please see the notes in the attached file

Author response: Corrections have been made as per the notes mentioned in the manuscript

Reviewer 3 Report

It's an excellent work for fulfilling recent demand of research for adverse environmental conditions. Now scientific world demanding such environmental friendly approach for increasing crop production with increasing stress tolerance. 

In my thought, authors were careless in checking their final version before submitting, because there are lots of silly mistake including spacing and so on.

For example, keywords arrangement should be in consistence.

dSm-1 should be dS m-1.

Be consistence in writing Na+ and Cl-  

revise the line of "Multiple genes are confluent in the salt tolerance mechanism confluent."

Scientific name must have to be in italic, please check the whole manuscript properly to fix all such issues.

There are plenty of issues need to be fixed in reference citation, like [112] documented, how? please be careful such things.   

Check the Figure 3 for fixing superscript and subscript issues and also check all others figures.

Except starting of sentence, there is no need for capitalized of Halotolerant.

don't give break in middle of the sentence under first paragraph of 4.1.1.

Tomatoes should be tomato, can you please explain why it would be tomatoes?

In case of abbreviation, all abbreviated word should be written in full form at first time appearance. Like EPS etc.

in table, plenty of spacing issue, please fix these carefully and be consistence in writing scientific name, because authors sometimes use abbreviated genus name and sometimes not. My suggestions will be give full name at first time and then use abbreviated one later. 

Be unique in writing unit of salinity in tables.

[133-134, 140, 127] reference number arrangement should be chronological in case of citation.

why [under abiotic stress] kept within third bracket?

Rewrite "6. Molecular understanding of PGPB mediated salt tolerance in plants" as present version is less emphasized. 

My another suggestion for authors will be please add some information about comparison between PGPR and others phytoprotectant approaches with recommendation of better one for using in increasing plant salinity tolerance.

Also add how PGPR works in modulating toxic Na+ uptake for making plants tolerant to ionic stress.    

Reviewer 2 Report

It's an excellent work for fulfilling recent demand of research for adverse environmental conditions. Now scientific world demanding such environmental friendly approach for increasing crop production with increasing stress tolerance. 

  • In my thought, authors were careless in checking their final version before submitting, because there are lots of silly mistake including spacing and so on.
  •  

Author response:  We are thankful to the reviewer for the excellent evaluation. All the suggestions of are well taken and the MSS has been revised as per the suggestions. All formatting issues have been taken care

  • For example, keywords arrangement should be in consistence.

Author response: Keywords are organized homogeneously

  • dSm-1 should be dS m-1.

Author response: Corrected

  • Be consistence in writing Na+and Cl-  

Author response:  Na+ and Cl- are now consistently mentioned throughout the manuscript.

  • revise the line of "Multiple genes are confluent in the salt tolerance mechanism confluent."

Author response: Multiple genes found associated with the salt tolerance mechanism. Line 150

  • Scientific name must have to be in italic, please check the whole manuscript properly to fix all such issues.

Author response: All scientific names are now written in italics

  • There are plenty of issues need to be fixed in reference citation, like [112] documented, how? please be careful such things.   

Author response:  Agreed and corrected as TrParray et al. [112]. Line 229

  • Check the Figure 3 for fixing superscript and subscript issues and also check all others figures.

Author response: Agreed and all errors in the figures are taken care.

  • Except starting of sentence, there is no need for capitalized of Halotolerant.

Author response: we follow the reviewer suggestion and revised the word Halotolerant as halotolerant.

  • don't give break in middle of the sentence under first paragraph of 4.1.1.

Author response:  Agreed and corrected

  • Tomatoes should be tomato, can you please explain why it would be tomatoes?

Author response: Agreed and revised as tomato. Line 259

  • In case of abbreviation, all abbreviated word should be written in full form at first time appearance. Like EPS etc.

Author response:  Agreed and corrected as per the suggestion

  • in table, plenty of spacing issue, please fix these carefully and be consistence in writing scientific name, because authors sometimes use abbreviated genus name and sometimes not. My suggestions will be give full name at first time and then use abbreviated one later. 

Author response: Spacing issues in Table are resolved now.

  • Be unique in writing unit of salinity in tables.

Author response: Agreed and corrected

  • [133-134, 140, 127] reference number arrangement should be chronological in case of citation.

Author response:  Corrected.

  • why [under abiotic stress] kept within third bracket?

Author response: bracket has removed

  • Rewrite "6. Molecular understanding of PGPB mediated salt tolerance in plants" as present version is less emphasized. 

Author response: we follow the reviewer suggestion and elaborate section 6. Molecular understanding of PGPB mediated salt tolerance in plants.

Round 2

Reviewer 3 Report

There are still many spacing and formatting mistake, like superscript specially in revised portion.